# The Role of Decompressive Craniectomy on Functional Outcome, Mortality and Seizure Onset after Traumatic Brain Injury

**DOI:** 10.3390/brainsci13040581

**Published:** 2023-03-29

**Authors:** Valeria Pingue, Valentina Boetto, Anna Bassetto, Maruska Nava, Antonio Nardone, Chiara Mele

**Affiliations:** 1Istituti Clinici Scientifici Maugeri IRCCS, Neurorehabilitation and Spinal Unit of Pavia Institute, 27100 Pavia, Italy; 2Department of Clinical-Surgical, Diagnostic and Pediatric Sciences, University of Pavia, 27100 Pavia, Italy

**Keywords:** decompressive craniectomy, neurorehabilitation, outcome, traumatic brain injury, seizures

## Abstract

Background: Decompressive craniectomy (DC) to treat increased intracranial pressure after a traumatic brain injury (TBI) is a common but controversial choice in clinical practice. This study aimed to determine the impact of DC on functional outcomes, mortality and the occurrence of seizures in a large cohort of patients with TBI. Methods: This retrospective study included patients with TBI consecutively admitted for a 6-month neurorehabilitation program between 1 January 2009 and 31 December 2018. The radiological characteristics of brain injury were determined with the Marshall computed tomographic classification. The neurological status and rehabilitation outcome were assessed using the Glasgow Coma Scale (GCS) and the Functional Independence Measure (FIM), which were both assessed at baseline and on discharge. Furthermore, the GCS was recorded on arrival at the emergency department. The DC procedure, prophylactic antiepileptic drug (AED) use, the occurrence of early or late seizures (US, unprovoked seizures) and death during hospitalization were also recorded. Results: In our cohort of 309 adults with mild-to-severe TBI, DC was performed in 98 (31.7%) patients. As expected, a craniectomy was more frequently performed in patients with severe TBI (*p* < 0.0001). However, after adjusting for the confounding variables including GCS scores, age and the radiological characteristics of brain injury, there was no association between DC and poor functional outcomes or mortality during the inpatient rehabilitation period. In our cohort, the independent predictors of an unfavorable outcome at discharge were the occurrence of US (β = −0.14, *p* = 0.020), older age (β = −0.13, *p* = 0.030) and the TBI severity on admission (β = −0.25, *p* = 0.002). Finally, DC (OR 3.431, 95% CI 1.233–9.542, *p* = 0.018) and early seizures (OR = 3.204, 95% CI 1.176–8.734, *p* = 0.023) emerged as the major risk factors for US, independently from the severity of the brain injury and the prescription of a primary prophylactic therapy with AEDs. Conclusions: DC after TBI represents an independent risk factor for US, regardless of the prescription of prophylactic AEDs. Meanwhile, there is no significant association between DC and mortality, or a poor functional outcome during the inpatient rehabilitation period.

## 1. Introduction

Traumatic brain injury (TBI) is a recognized public health problem and represents a leading cause of disability and mortality worldwide [1,2]. TBI is a direct consequence of an external impact force, which leads to transient or permanent damage to the central nervous system (CNS) [3,4]. Based on the dynamics of the damage, TBI can be classified into a closed-head injury, when the brain tissue is damaged but the skull remains intact, or a penetrating head injury when the skull is damaged [5].

It is known that multiple pathophysiological mechanisms contribute to determine several cognitive, behavioral and functional alterations in TBI [4,6]. The primary injury related to mechanical damage can induce an increase in intracranial pressure (ICP) and promotes a complex cascade of biochemical, inflammatory and metabolic alterations leading to secondary injury, which has been related to the onset of clinically relevant neurological complications [7]. Decompressive craniectomy (DC) represents a useful neurosurgical procedure to counteract the TBI-related increase of ICP and to improve the brain oxygen delivery, partially preventing the aforementioned secondary damage [8]. While some authors have demonstrated that DC is associated with a significant decrease in the mortality rate in TBI patients with refractory high ICP [9], several studies have suggested that DC could contribute to altering the cerebrospinal fluid circulation, cerebral blood flow and glucose metabolism [10,11]. Recently, it has been hypothesized that these alterations could negatively affect the neurological and functional outcomes, although the clinical evidence is still controversial [12].

Post-traumatic seizures (PTS) represent one of the most prevalent neurological complications of TBI and have been documented in a varying proportion of patients who underwent DC [13]. It is worth mentioning that most of the patients undergoing DC fall into the severe brain injury category, which is characterized by a higher risk of seizure onset per se [14,15]. PTS have been classified as acute symptomatic seizures (ASS) when they occur within one week of the injury and are related to a transitory decrease in the seizure threshold [16,17], and unprovoked seizures (US) when they occur after one week of the injury and are associated with structural changes in the neuronal networks triggered by the secondary injury cascade [17]. Prophylactic antiepileptic drug (AED) therapy is recommended within the first week after trauma to prevent ASS, which are considered a potential risk factor for US [18,19]. The need for the prolonged used of prophylactic AEDs is still controversial [20,21,22], as several authors have suggested that the overuse of AEDs for prophylaxis is not effective in reducing the incidence of US and is associated with cognitive and behavioral complications, thus negatively influencing the rehabilitation outcome [23,24,25,26]. Although there is sufficient literature supporting the short-term benefits of AEDs in post-TBI subjects, there has been controversy about the time frame for continuing AED therapy in patients undergoing DC for more severe TBI [27].

With the aim of evaluating the role of primary DC in influencing the long-term functional outcomes, mortality and seizure onset, we conducted a retrospective longitudinal study on a large cohort of rehabilitation patients followed from the post-acute phase up to six months after TBI.

## 2. Materials and Methods

### 2.1. Study Design and Population

This observational retrospective cohort study enrolled 309 patients with traumatic brain injury (TBI), consecutively admitted to the Neurorehabilitation Unit of ICS Maugeri of Pavia, Italy between 1 January 2009 and 31 December 2019. The eligibility criteria included: (1) age  ≥18 years; (2) a diagnosis of TBI; (3) admission to a hospital emergency unit within 24 h of the traumatic event; (4) primary DC performed in the acute care setting as a part of their neurocritical management; (5) admission to our rehabilitation unit for an intensive neurorehabilitation program within one week of DC to continue the clinical care and rehabilitation programs started at the acute care units of the province of Pavia. Individuals were excluded from the study if data concerning their acute care were not available. We also excluded patients with pre-existing neurological events or diseases. The study design conformed to the ethical guidelines of the Declaration of Helsinki and was approved by the local Ethical Committee ICS Maugeri (#2214 CE). The participants or authorized representatives signed a written informed consent form.

### 2.2. Variables, Data Sources and Measurements

The data were retrieved from the electronic hospital records at baseline and on discharge, and included the following variables: sex, age at the occurrence of injury, the presence of subarachnoid hemorrhage, cerebral edema, intraparenchymal hematoma, associated extra-cranial traumatic complications (such as thoracic trauma, cardiac and kidney failure), the occurrence of seizures, the prescription of prophylactic AEDs, neurological and functional assessments, death during rehabilitation. We also determined the radiological characteristics of the brain injury with the Marshall computed tomographic (CT) classification [28] that categorizes injuries into six classes based on: the degree of swelling as determined by basal cistern compression and midline shift and the presence and size of focal lesions, depending on whether the lesion volume exceeded 25 cm^3^.

All the participants underwent an inpatient neurorehabilitation program consisting of individual 3-hour daily treatment cycles, 6 days per week inclusive of physiotherapy, occupational therapy, speech therapy, cognitive training and nutrition assistance, as well as psychological and social support.

### 2.3. Seizures and Antiepileptic Drugs

In accordance with the ILAE criteria [17], seizures occurring during the acute and rehabilitation period were classified into two categories defined by taking into account the time elapsed from the occurrence of the brain injury: ASS, if occurring within 7 days of TBI; late, US, if occurring >7 days after TBI. We excluded any provoked seizures after the acute phase. Any paroxysmal event that occurred during hospitalization, either described by the patients or eye-witnessed, was examined by clinicians. Epileptic seizures were diagnosed on the basis of clinical features and EEG findings. AEDs were prescribed in the acute setting care as the primary prophylaxis, or in both the acute and rehabilitation settings after the first occurrence of seizures (secondary prophylaxis). AEDs were further subdivided into first- or second-generation drugs [29].

### 2.4. Neurological and Functional Assessment

The Glasgow Coma Scale (GCS) and Functional Independence Measure (FIM) scale were administered on admission (T0) and at discharge (T1) to evaluate the neurological and rehabilitation outcomes, respectively. Furthermore, the GCS was recorded on arrival at the emergency department (GCS on Arrival; GCSoA). The GCS represents a standardized system for assessing the degree of neurological impairment and to identify the seriousness of the injury in relation to the outcome. The GCS assessment includes three determinants: eye opening, verbal responses and motor response or movement. These determinants are evaluated separately according to a numerical value which indicates the level of consciousness and the degree of neurological dysfunction. The total score ranges from 15 to 3. Patients are considered to have experienced a “mild” brain injury when their score is between 13 and 15. A score between 9 and 12 indicates a “moderate” brain injury, and a score equal to 8 or less reflects a “severe” brain injury [30]. The rehabilitation outcomes were evaluated through the FIM scale, an 18-item measurement tool that explores an individual’s physical, psychological and social function [31,32]. The tool is used to assess the patient’s level of disability as well as any change in patient status in response to rehabilitation or medical interventions [33]. ΔFIM scores corresponding to T1 minus T0 values were also calculated.

### 2.5. Statistical Analysis

The values are expressed as the median and interquartile range (IQR) or an absolute number and percentage. The data were tested for normality of distribution with the Shapiro–Wilk test. The Mann–Whitney U and chi-square tests were used for comparisons between the groups. A multivariable logistic regression analysis was used to identify the independent risk factors of seizure occurrence and mortality during rehabilitation. The odds ratio (OR), 95% confidence interval (95% CI), and related significance are reported. A multiple linear regression analysis was used to evaluate the independent predictors of rehabilitation outcome. The multilinear model included FIM T1 or ΔFIM as dependent variables and cranioplasty, sex, age and brain injury characteristics as independent variables. The β coefficients and significance obtained from the models were reported. A value of *p*  <  0.05 was considered as statistically significant. The statistical analyses were performed using SPSS version 21 (IBM Corporation, Somers, NY, USA).

## 3. Results

### 3.1. Clinical and Functional Characteristics

The cohort included 309 adult patients with mild-to-severe TBI (Figure 1).

A summary of the clinical and functional characteristics of the whole population is reported in Table 1. Overall, 78.3% were males (male to female ratio, 3.6:1). More than half of the patients were over 65 years of age at the time of injury. The GCS administered on arrival at the emergency department assessed severe TBI in 67.2% of cases, moderate in 21.5% and mild in 11.3%. Whereas, based on the GCS scores on admission, TBI was classified as mild in 36.5% of cases, moderate in 40.8% and severe in 22.6%. As a direct consequence of TBI, subarachnoid hemorrhage was detected in 38.6% of patients, cerebral edema in 17.6% and intraparenchymal hematoma in 55.6%. As a likely consequence of the injury, seizures occurred in 63 patients (20.4%). ASS were documented in 22 cases (7.1%) and US in 32 (10.4%), whereas nine patients (2.9%) first presented with ASS and then US. Primary prophylactic therapy with AEDs was started in 23.3% of cases. Among the AEDs, second-generation drugs were the most frequently used for primary prophylaxis (76.4% of cases), and, in particular, levetiracetam was prescribed to 50/55 patients. The first-generation AEDs were valproic acid (7/17 patients), carbamazepine (5/17) and phenytoin (5/17). Overall, second-generation AEDs were the most prescribed in patients who underwent DC (*p* = 0.003). Regarding neurosurgical interventions, 31.7% of patients underwent DC in the acute phase of brain injury (hemicraniectomy in 70/98 patients; bifrontal craniectomy in 18/98). On admission to our unit, extra-cranial traumatic complications after TBI were diagnosed in 59.2% of cases. The most frequently reported were thoracic trauma (55/183) and orthopedic and/or vascular trauma of the limbs (40/183). Death during rehabilitation was documented in 41 patients (13.3%).

Comparison analyses were conducted between patients who underwent DC and those who did not (Table 1). As expected, a craniectomy was more frequently performed in patients with severe TBI on arrival at the emergency department (79.5% vs. 60.4%, *p* = 0.002) and those with the worst CT scan findings based on the Marshall classification (*p* < 0.0001). Although AEDs as a primary prophylaxis were more frequently prescribed in patients undergoing DC (35.7% vs. 17.5%, *p* < 0.0001), a higher prevalence of US was documented in these patients than their counterparts (17.3% vs. 7.1%, *p* = 0.018). As regards mortality within 6 months of brain injury, the patients who underwent DC had a higher mortality rate than their counterparts (23.5% vs. 8.5%, *p* = 0.026). No significant differences were found between the two groups in terms of sex, age at diagnosis and the characteristics of brain damage.

### 3.2. Functional Outcome

A multiple linear regression analysis was conducted to evaluate the predictive role of craniectomy on the functional outcome. The models achieving the highest coefficient of determination (R^2^) are reported in Table 2. US emerged as an independent predictor of a worse functional outcome in terms of the FIM total score at discharge (FIM T1) (β = −0.14, *p* = 0.020) and the FIM variation (ΔFIM) during a rehabilitation hospital care program (β = −0.20, *p* = 0.007). Instead, there was no association between DC and mortality, and poor functional outcome. As expected, the neurological and functional status at T0 and an age of over 65 years emerged as significant predictors of the functional outcome independently from the other variables included in the regression model.

### 3.3. Mortality

A multivariable logistic regression analysis was conducted to evaluate the potential risk factors for mortality during the rehabilitation period (Table 3). An age of over 65 years (OR = 6.185, 95% CI 2.464–15.526, *p* < 0.0001) and the GCS score on admission to the neurorehabilitation unit (OR = 2.648, CI 95% 1.360–5.155, *p* = 0.004) emerged as the main independent risk factors for mortality. Contrariwise, neurosurgical procedures, the occurrence of seizures and primary prophylactic AEDs did not influence the mortality rate in this context.

### 3.4. Seizures and Primary Prophylactic AED Therapy

A multivariable logistic regression analysis was conducted to identify the potential risk factors for seizure onset within the 6-month inpatient rehabilitation period (Table 4). DC emerged as the main risk factor for US onset independently from sex, age, GCS score, CT scan findings based on the Marshall classification and the prescription of prophylactic AEDs (OR 3.431, 95% CI 1.233–9.542, *p* = 0.018). As expected, another important risk factor for US onset was the occurrence of ASS (OR = 3.204, 95% CI 1.176–8.734, *p* = 0.023). No risk factors were identified for ASS onset in our TBI cohort. Of note, the administration of a primary prophylactic therapy with AEDs was not associated with a lower prevalence of seizures.

## 4. Discussion

The aim of our study was to investigate the role of primary DC in influencing functional outcomes, mortality and seizure onset in a large cohort of post-TBI rehabilitation patients. We monitored our patients from the acute phase to discharge from our rehabilitation unit up to 6 months after injury. In this time frame lesions tend to stabilize and patients can achieve the maximum functional recovery, even if the risk of seizure occurrence is high [34]. Our main findings indicate that DC in this critical period represents a risk factor for US independently from sex, age, TBI severity on admission and the prescription of prophylactic AEDs. Contrariwise, there was no association between DC and mortality or DC and a poor functional outcome during the inpatient rehabilitation program. Treatment for ICP elevation is mandatory for the management of patients with severe TBI [35]. In this context, DC represents a cornerstone of the neurosurgeon’s choices for treating ICP elevation after brain damage [36]. However, this neurosurgical procedure remains a controversial topic in the TBI field, in particular concerning the relationship between DC, neurological complications and functional outcome in the course of rehabilitation [35]. In fact, it is difficult to accurately predict the long-term global outcomes of patients with brain injury, despite the important advances in identifying the early-stage prognostic factors of TBI [37]. For this reason, in the last decade, the effects of DC on neurological and functional recovery have been increasingly examined in observational studies and clinical trials [27,38,39,40,41]. Williams et al. found that DC resulted in a good functional outcome in > 50% of patients with severe TBI [41]. The greatest benefit was observed in younger patients with a demonstrable reduction in ICP after decompression. On the contrary, Cooper in an observational study and Hutchinson in a clinical trial observed that the DC procedure was associated with a significant worsening in the functional outcome compared to control groups [39,40]. Likewise, the most recent clinical trials have suggested that DC after severe diffuse TBI and early refractory intracranial hypertension increases the incidence of vegetative survivors and does not improve outcomes [38]. The difference between findings could be due to the heterogeneity of populations and settings as well as the lack of adjustment for important covariates that limit the interpretation of the results. In our cohort, DC was performed in patients with more severe TBI characteristics at the baseline, as expected. However, after adjusting for the confounding variables, including GCS scores, age and radiological findings based on the Marshall classification, there was no association between DC and a worse functional outcome or mortality during the inpatient rehabilitation period. Overall, the occurrence of US emerged as an independent predictor of a worse functional outcome after the rehabilitation program. Whether US and secondary epilepsy depend on the severity of brain damage, which, in turn, drives poor outcomes, remains an open question [21]. Altogether, seizures and brain damage could act synergistically to hamper the recovery and drive poor outcomes. The other main predictors of a poor functional outcome and mortality were older age and poor neurologic and functional assessment on admission, independently from the potential detrimental role of the neurosurgical procedure. All these findings could be explained by considering the extreme heterogeneous pathophysiological TBI mechanisms, which act synergistically, contributing to the impairment of neurological and functional outcomes after a mild-to-severe injury [4,6]. The mechanical damage promotes a cascade of metabolic, biochemical and inflammatory alterations leading to long-term TBI complications, including ischemic cell damage, seizure and death [42]. Thus, it is plausible that, in itself, TBI severity is the major determinant of patient outcome, regardless of the surgical procedure performed. Regarding seizure onset, DC and ASS emerged as the main risk factors for US during the rehabilitation period, independently from prophylactic AED prescription. In this context, DC is effective in reducing the incidence of cerebral edema and intracranial pressure, but at the same time it is not effective in improving post-TBI related pathology, such as epilepsy. Indeed, US are an expression of the structural damage of neural networks and connectivity following a brain injury [43]. To date, no pharmacological interventions, including AEDs, can effectively prevent the development of late seizures and/or epilepsy after potential epileptogenic insults [24]. Therefore, the current guidelines recommend the use of AEDs for seizure prophylaxis only during the first weeks after TBI [44]. By inference, short-term prophylaxis with AEDs could be appropriate in patients scheduled for neurosurgery after TBI. However, our data did not allow us to draw conclusions on the efficacy of AEDs in these patients, and prospective and randomized controlled trials are needed to address this topic. This study has some limitations, which should be pointed out. The study design does not allow us to draw any conclusions about the mechanisms involved in the relationship between a craniectomy and functional outcomes in TBI. Moreover, the retrospective design involves the review of charts not originally designed for collecting data for research, with a lack of in-depth analysis of the multiple factors that contribute to GCS scores and head injury severity. For instance, data on the standard criteria for patients that receive a DC were unavailable. Further, in this context, the choice to perform a DC was made by the neurosurgeon, probably taking into account the clinical severity. In spite of these limitations, our study included a large sample size and an extensive characterization of our populations. Moreover, we resolved this potential bias by weighting the regression analysis for the severity of damage and radiological findings, thus demonstrating the independent association between DC and rehabilitation outcomes.

## 5. Conclusions

In conclusion, our study suggests that DC represents an independent risk factor for US after TBI, in addition to other well-known factors, such as early seizures. Further, the prophylactic prescription of antiepileptic drugs appears to be ineffective in US prevention. Contrariwise, there was no association between DC and mortality, or DC and poor functional outcome during the inpatient rehabilitation period.

## Figures and Tables

**Figure 1 brainsci-13-00581-f001:**
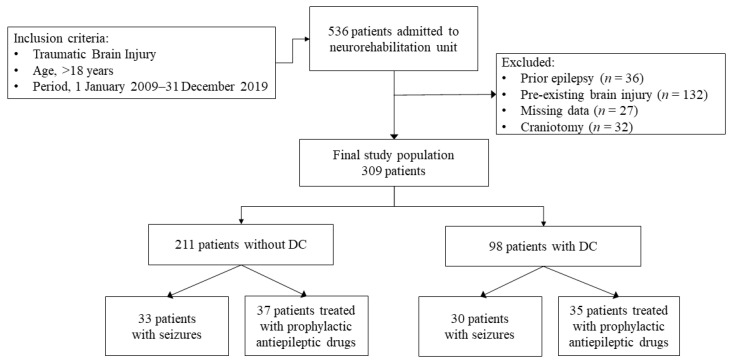
Traumatic brain injury patients flow diagram.

**Table 1 brainsci-13-00581-t001:** Clinical and functional characteristics of the population as a whole and subdivided into two groups based on whether or not craniectomy surgery was performed.

Variables		Whole Population (*n* = 309)	DC		*p*
			No (*n* = 211, 68.3%)	Yes (*n* = 98, 31.7%)	
		*n* (%)	*n* (%)	*n* (%)	
Age (years)	≤65	183 (59.2)	126 (59.7)	57 (58.2)	0.805
	>65	126 (40.8)	85 (40.3)	41 (41.8)	
Sex	Male	242 (78.3)	171 (81.0)	71 (72.4)	0.103
	Female	67 (21.7)	40 (20.0)	27 (27.6)	
Subarachnoid hemorrhage		121 (39.1)	88 (41.8)	33 (33.7)	0.060
Cerebral edema		53 (17.6)	34 (16.11)	19 (19.4)	0.517
Intraparenchymal hematoma		172 (55.7)	99 (46.9)	73 (74.5)	**<0.0001**
Extracranial traumatic complications		183 (59.2)	118 (55.9)	65 (66.3)	0.106
Patients with seizures		63 (20.4)	33 (15.6)	30 (30.6)	**0.016**
Type of seizures	ASS	22 (7.1)	14 (6.6)	8 (8.2)	0.894
	US	32 (10.4)	15 (7.1)	17 (17.3)	**0.018**
	ASS + US	9 (2.9)	4 (1.9)	5 (5.1)	**0.047**
Prophylaxis with AEDs	I generation	17 (5.5)	9 (4.3)	8 (8.7)	0.183
	II generation	55 (17.8)	28 (13.3)	27 (27.6)	**0.003**
Adapted Marshall classification (Data available for 289 patients)	Diffuse injury I	97 (31.4)	91 (43.1)	1 (1.0)	**<0.0001**
	Diffuse injury II	53 (17.1)	48 (22.7)	2 (2.0)	**<0.0001**
	Diffuse injury III (swelling)	61 (19.7)	44 (20.8)	17 (17.3)	0.540
	Diffuse injury IV (shift)	78 (25.2)	10 (4.7)	68 (69.4)	**<0.0001**
	Evacuated lesion	0 (0.0)	0 (0.0)	8 (8.2)	**<0.0001**
	Non evacuated lesion	0 (0.0)	0 (0.0)	0 (0.0)	n.a.
GCSoA (Data available for 247 patients)	Mild	28 (11.3)	24 (15.1)	4 (4.5)	**0.011**
	Moderate	53 (21.5)	39 (24.5)	14 (15.9)	0.145
	Severe	166 (67.2)	96 (60.4)	70 (79.5)	**0.002**
GCS T0	Mild	113 (36.5)	91 (43.1)	22 (22.5)	**<0.0001**
	Moderate	126 (40.8)	88 (41.8)	38 (38.8)	0.488
	Severe	70 (22.6)	31 (14.7)	39 (39.8)	**<0.0001**
GCS T1	Mild	200 (74.4)	158 (81.9)	42 (55.3)	**<0.0001**
	Moderate	41 (15.2)	23 (11.9)	18 (23.7)	**0.02**
	Severe	28 (10.4)	12 (6.2)	16 (21.0)	**0.0003**
FIM T0 (median (IQR)) total score		19 (18–56)	26 (18–65)	18 (18–22)	**0.001**
FIM T1 (median (IQR)) total score		82 (23–117)	104 (40–122)	28 (18–91)	**<0.0001**
Mortality within 6 months		41 (13.3)	18 (8.5)	23 (23.5)	**0.026**

Data are expressed as the median and interquartile range (IQR) or absolute number and percentage. The comparisons between groups were performed with the χ^2^ or Mann–Whitney U tests. Significant difference are shown in bold characters. AEDs, antiepileptic drugs; ASS, acute symptomatic seizures; DC, decompressive craniectomy; FIM, functional independence measure; GCS, Glasgow Coma Scale; GCSoA, Glasgow Coma Scale on Arrival; T0, admission; T1, discharge; US, unprovoked seizures.

**Table 2 brainsci-13-00581-t002:** Multiple linear regression models to evaluate the potential predictors of functional outcome, in terms of FIM on discharge (FIM T1) and FIM variation during rehabilitation (ΔFIM).

Regression Model	FIM T1 (R^2^ = 0.542)		ΔFIM (R^2^ = 0.250)	
Independent Variables	Beta	*p*-Value	Beta	*p*-Value
Sex (M = 0, F = 1)	0.06	0.306	0.07	0.296
Age >65 years	−0.13	0.030	−0.17	**0.028**
Adapted Marshall classification	−0.14	0.055	−0.20	**0.031**
Glasgow Coma Scale on arrival (mild = 1, moderate = 2, severe = 3)	−0.06	0.340	−0.08	0.319
Glasgow Coma Scale on admission (mild = 1, moderate = 2, severe = 3)	−0.25	0.002	−0.30	**0.003**
FIM total score on admission	0.37	<0.0001	−0.36	**<0.0001**
Extracranial traumatic complications	−0.04	0.526	−0.07	0.360
Decompressive craniectomy	−0.08	0.262	−0.09	0.310
Acute symptomatic seizures	−0.06	0.286	−0.07	0.310
Unprovoked seizures	−0.14	0.020	−0.20	**0.007**
I generation prophylactic AEDs	−0.01	0.855	−0.01	0.857
II generation prophylactic AEDs	−0.01	0.832	−0.02	0.784

Significant difference are shown in bold characters. AEDs, antiepileptic drugs; FIM, functional independence measure.

**Table 3 brainsci-13-00581-t003:** Multivariate logistic regression analysis showing the potential risk factors for mortality within 6 months of brain injury.

Regression Model	Death during Rehabilitation (Dependent Variable) (No = 0, Yes = 1)		
Independent Variables	OR	95% CI	*p*-Value
Sex (M = 0, F = 1)	0.904	0.345–2.634	0.836
Age >65 years	6.185	2.464–15.526	**<0.0001**
Adapted Marshall classification	1.258	0.787–2.013	0.338
GCS on arrival (mild = 1, moderate = 2, severe = 3)	0.804	0.399–1.622	0.543
GCS on admission (mild = 0, moderate = 1, severe = 2)	2.648	1.360–5.155	**0.004**
Extracranial traumatic complications	0.626	0.249–1.572	0.319
Decompressive craniectomy	2.046	0.663–6.311	0.213
Acute symptomatic seizures	0.721	0.157–3.319	0.674
Unprovoked seizures	0.502	0.148–1.700	0.268
Prophylaxis with AEDs	0.611	0.232–1.612	0.320

Significant difference are shown in bold characters. AEDs, antiepileptic drugs; GCS, Glasgow Coma Scale.

**Table 4 brainsci-13-00581-t004:** Multivariable logistic regression analysis showing the potential risk factors for seizure occurrence in patients with TBI.

Regression Model	Acute Symptomatic Seizure			Unprovoked Seizures		
	OR	95% CI	*p* Values	OR	95% CI	*p* Values
Sex (M = 0, F = 1)	1.098	0.402–3.000	0.855	0.778	0.296–2.040	0.609
Age >65 years	1.235	0.493–3.090	0.652	0.872	0.381–1.995	0.746
Adapted Marshall classification	1.203	0.769–1.882	0.419	0.922	0.623–1.365	0.686
GCS on arrival (mild = 0, moderate = 1, severe = 2)	1.071	0.539–2.128	0.844	0.742	0.397–1.389	0.351
GCS on admission (mild = 0, moderate = 1, severe = 2)	0.836	0.458–1.526	0.559	1.486	0.851–2.595	0.164
Decompressive craniectomy	1.594	0.511–4.977	0.422	3.431	1.233–9.542	**0.018**
ASS	n.a.	n.a.	n.a.	3.204	1.176–8.734	**0.023**
Prophylaxis with AEDs	0.392	0.090–1.154	0.082	0.907	0.373–2.205	0.830

Significant differences are shown in bold characters. AEDs, antiepileptic drugs; ASS, acute symptomatic seizure; GCS, Glasgow Coma Scale.

## Data Availability

The data presented in this study are available on request from the corresponding author.

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
