# Peer review of "The Role of Decompressive Craniectomy on Functional Outcome, Mortality and Seizure Onset after Traumatic Brain Injury"

_brainsci, 2023, doi:10.3390/brainsci13040581_

Round 1

Reviewer 1 Report

The ms by Pingue et al. touches on a very interesting and important topic the value of maximal treatment (i.e. decompressive craniectomy) after TBI.

That subject needs exploration and the article adds some information to that. The analyzed group is sufficiently numerous.

However, on the level of data gathering, it lacks many important details, that could potentially impact the results. The conclusion that DC after TBI is an independent risk factor for mortality is quite astonishing, especially in contrast to existing data that it decreases mortality (with increasing morbidity, of course, and poor outcome).

Nevertheless, the dataset being the base for the analysis and conclusion is not detailed enough. TBI is a very general term, and the patients can present a variety of traumas (edema, contusion, DAI, hematomas), and depending on the amount and quantity the prognosis is different. In the presented cohort there's no differentiation. SAH alone is a weird factor because it is probably found in nearly all TBIs.

GCS can initially classify the head trauma, but the patient can worsen very fast (within minutes) with edema or hematoma, therefore it says nothing about the severity of head trauma. Also, the time of GCS assessment is crucial - the prognostic one is the first GCS, that is taken by the emergency team on site (here, as I understood, it was taken at the admission to the hospital).

The presence of coexisting diseases should also be taken into account.

Moreover, the technique of DC is important (one-two-sided, bifrontal, was the hematoma evacuated?, was it big enough, were the patients analyzed after bone reimplantation?).

Summing that up, the severity of head trauma and initial patient status should be better characterized, as should the details about the procedure (DC). Also, more patient-related factors should be included (comorbidities, smoking, etc.).

Author Response

Answers to Reviewer 1’s criticisms:

We thank the Reviewer for the insightful criticisms which have greatly contributed to improve the overall quality of our manuscript. Specific comments:

1) The conclusion that DC after TBI is an independent risk factor for mortality is quite astonishing, especially in contrast to existing data that it decreases mortality (with increasing morbidity, of course, and poor outcome). Nevertheless, the dataset being the base for the analysis and conclusion is not detailed enough. TBI is a very general term, and the patients can present a variety of traumas (edema, contusion, DAI, hematomas), and depending on the amount and quantity the prognosis is different. In the presented cohort there's no differentiation. SAH alone is a weird factor because it is probably found in nearly all TBIs.

Response: Thanks a lot for this important question. The retrospective nature of the study implied a lack of in-depth analysis of the multiple factors that contribute to GCS scores and severity of head injury. Accordingly, we have reported this criticism in the limitations section (page 9, lines 291-294). However, as requested by both reviewers, we have provided additional data to better characterize TBI. In particular, regarding the description of brain damage, we have reported the presence of intraparenchymal hematoma and cerebral edema in the methods and results section (page 2, lines 92-93; page 4, lines 157-158, page 5 table 1; page 6, table 2; page 8, table 4)

2) GCS can initially classify the head trauma, but the patient can worsen very fast (within minutes) with edema or hematoma, therefore it says nothing about the severity of head trauma. Also, the time of GCS assessment is crucial - the prognostic one is the first GCS, that is taken by the emergency team on site (here, as I understood, it was taken at the admission to the hospital).

The presence of coexisting diseases should also be taken into account.

Response: The Reviewer raised a valid concern. We have reported the data concerning the GCS administered on arrival to the emergency department (GCSoA) and the presence of extracranial traumatic complications (page 2, line 93; page 3, line 118-119; page 4, lines 153-155; page 5, table 1; page 6, table 2; page 7, table 3; page 8, table 4).

 3) Moreover, the technique of DC is important (one-two-sided, bifrontal, was the hematoma evacuated?, was it big enough, were the patients analyzed after bone reimplantation?).

Summing that up, the severity of head trauma and initial patient status should be better characterized, as should the details about the procedure (DC). Also, more patient-related factors should be included (comorbidities, smoking, etc.).

Response: We have reported in the results section the description of DC technique (page 4, line 166-167). To better characterized the severity of the brain injury we reported a description of radiological severity with the Marshall scores (page 4, lines 174-175, page 5, table 1). The retrospective design of the study doesn’t allow us to report more details of the surgical procedure.  We added this issue as a limitation of the study (page 9, lines 293-294).

Reviewer 2 Report

This paper is not clinically relevant.  In active practice only patients with severe TBI get a craniectomy.... comparing outcomes to people who do not need a lifesaving surgery is not helpful.  

There have already been many published papers about outcome for craniectomy in severe TBI and it is accepted that the procedure is essentially lifesaving and functional outcome is dependent on the primary insult severity.  

Author Response

We thank the Reviewer 2’s for the insightful criticisms.

The main aim of our work was to better define the relationship between DC and functional outcome and neurologic complication on the rehabilitation course of brain injury. Indeed, although important advances to predict prognosis early after TBI, it remains hard to accurately predict outcomes for an individual patient over the long term. Among these patients, we also wanted to evaluate the development of seizures, and in turn, whether primary prophylactic AEDs could be effective in seizures prevention.

We think that this message could support clinicians in their decision-making process in daily practice

Reviewer 3 Report

Dear Editor,

in the current manuscript, Pingue et al describe the predictive role of decompressive craniectomy (DC) on functional outcome, mortality and seizure occurence (as divided into acute and late seizures). They describe a cohort of a total of 309 patients of whom 98 received a DC. All together they found that DC was an independent predictor of outcome at 6 months when evaluated using FIM. DC as well as early seizures were also predictors of late seizures. While they raise some interesting points there are some major concerns that need to be addressed:

Major:

1.     In general, even though the authors describe the clinical severity of disease, a description of radiological severity is lacking. Multiple widely validated scores (e.g. Marshall and Rotterdam CT scores) have been shown to be beneficial for outcome prediction. Alternatively, the Impact score could be used for the description of both clinical and radiological severity.

2.     Did any of the patients suffer extracranial traumatic sequelae?

3.     Outcome is evaluated using the GCS as well as the FIM. GCS is not a recognised outcome descriptor. I would suggest to use GOS or GOSE.

4.     Is DC available at all time? Are there any standard criteria for patients that receive a DC at this institution? Where all the DCs unilateral? Is there data on the adequacy of size or benefit of DC in these patients?

5.     A large portion of patients might have covert seizures in particular in cases with severe TBI in need of DC. What are the local standard operating procedures in regards to performance of EEG and cEEG?

6.     Were provoked seizures after 7 days excluded from the late seizures?

7.     Currently instead of GCS, AIS over 3 is more commonly used to evaluate severity of TBI as the severity can be vastly underestimated or overestimated by GCS alone.

8.     In 23.3% of patients primary prophylactic AED was started. Please explain the reasons in which these are started (e.g. penetrating injury? Severity of injury? Acute seizures?). Please also describe which AED is used as some have shown to impair outcome on a long-term basis.

9.     What is FIM variation? Did you evaluate it multiple times during the rehabilitation or is it just the difference between admission and discharge FIM?

Author Response

Answers to Reviewer 3’s criticisms:

We thank the Reviewer for the insightful criticisms which have greatly contributed to improve the overall quality of our manuscript. Specific comments:

  1. In general, even though the authors describe the clinical severity of disease, a description of radiological severity is lacking. Multiple widely validated scores (e.g. Marshall and Rotterdam CT scores) have been shown to be beneficial for outcome prediction. Alternatively, the Impact score could be used for the description of both clinical and radiological severity.

Response: The Reviewer raised a valid concern. We have reported in the results section the description of radiological severity with the Marshall scores (page 4, lines 174-175, page 5, table 1).

  1. Did any of the patients suffer extracranial traumatic sequelae?

Response: The Reviewer raised a valid concern. We have introduced into our analyses data concerning the presence on admission to our unit of extracranial traumatic complications (page 2, line 93; page 3, line 118-119; page 4, lines 153-155; page 5, table 1; page 6, table 2; page 7, table 3; page 8, table 4).

  1. Outcome is evaluated using the GCS as well as the FIM. GCS is not a recognised outcome descriptor. I would suggest to use GOS or GOSE.

This is a good point. The main purpose of the study was the evaluation of DC on the rehabilitation course and the FIM scale that we used is a valid tool to measure this outcome. However, the retrospective design of the study doesn’t allow us to report more details of the patients outcome.  We added this issue as a limitation of the study (page 9, lines 291-293). 

  1. Is DC available at all time? Are there any standard criteria for patients that receive a DC at this institution? Where all the DCs unilateral? Is there data on the adequacy of size or benefit of DC in these patients?

Response: We thank the Reviewer. We have reported in the results section the description of DC procedure (page 4, line 166-167). But the point raised by the reviewer deserves to be included among the limitations of the study, so we have added the following sentence in the Discussion: ‘For instance, data on standard criteria for patients that receive a DC, were unavailable.’ (page 9, lines 293-294).

  1. A large portion of patients might have covert seizures in particular in cases with severe TBI in need of DC. What are the local standard operating procedures in regards to performance of EEG and cEEG?

Response: We agree with the reviewer, so we have added a sentence that properly defines how we managed seizures in the Methods: ‘Any paroxysmal event that occurred during hospitalization, either described by patients, or eye witnessed, was examined by clinicians. Epileptic seizures were diagnosed on the basis of clinical features and EEG findings.’ (page 3, lines 108-110).

  1. Were provoked seizures after 7 days excluded from the late seizures?

Response: This is a good point. We have excluded any provoked seizures occurred after the acute phase. We have added the information in the Methods (page 3, line 108).

  1. Currently instead of GCS, AIS over 3 is more commonly used to evaluate severity of TBI as the severity can be vastly underestimated or overestimated by GCS alone.

This is a good point. However, given the retrospective nature of the study, we used the available data. We added this issue as a limitation of the study (page 9, lines 291-293).

  1. In 23.3% of patients primary prophylactic AED was started. Please explain the reasons in which these are started (e.g. penetrating injury? Severity of injury? Acute seizures?). Please also describe which AED is used as some have shown to impair outcome on a long-term basis.

Response: We agree with the reviewer, and have added a sentence that properly explain the generation and type of AEDs used as prophylactic therapy, in the methods and in the results (page 3, line 113; page 4, lines 161-164). As regard the criteria of prescription, there were no standardized protocols for the prescription of prophylactic AED therapy.

Therefore, in this setting, AEDs were prescribed at clinician’s discretion, likely taking into account the severity of the clinical and radiological picture and the need of neurosurgical procedure.

  1. What is FIM variation? Did you evaluate it multiple times during the rehabilitation or is it just the difference between admission and discharge FIM?

Response: We thank the Reviewer, and have added a sentence that properly defines FIM variation, in the Methods: “ΔFIM scores, corresponding to T1 minus T0 values, were also calculated” (page 3, line 130-131).

Round 2

Reviewer 3 Report

I much appreciate the effort of the authors in improving this manuscript. They have answered most of my concers. However, I would still like to address the following concern:

1. I would suggest to add the Marshall Score to the prediction models of outcome and seizures instead of the seperate imaging features to prove independence of the other factors on outcome

Author Response

Answers to Reviewer criticisms:

We warmly thank the Reviewer for this suggestion, which certainly broadens our discussion on the case and improve the overall quality of our manuscript. Specific comments:

  • I would suggest to add the Marshall Score to the prediction models of outcome and seizures instead of the seperate imaging features to prove independence of the other factors on outcome.

Response: We have reported the Marshall scores as indipendendent variable in the prediction models of outcome and seizures (page 6, table 2; page 7, table 3; page7 table 4) .Repeating the analyses, we found no significant association between DC and mortality and poor functional outcomes during the inpatient rehabilitation period. Whilst, the neurosurgery procedure remains as a risck factor for US, independently from AEDs prescription.

Consequently, we rephrased the abstract, the results and the discussion.
